# Enhancing nutritional value and quality of cookies through pumpkin peel and seed powder fortification

Md. Asaduzzaman[1,2,3☯], Mst. Sorifa Akter[4☯]*, Mahbubur Rahman[2]

1 Department of Food Processing and Preservation, Hajee Mohammad Danesh Science and Technology University, Dinajpur, Bangladesh, 2 Department of Food Engineering and Nutrition Science, State University of Bangladesh, Dhaka, Bangladesh, 3 School of Food Science and Technology, Jiangnan University, Wuxi, China, 4 Department of Nutrition and Food Engineering, Daffodil International University, Dhaka, Bangladesh

☯ These authors contributed equally to this work.
* sorifa.nfe@diu.edu.bd

**Data Availability Statement:** All relevant data are within the manuscript and its Supporting Information files.

**Funding:** This study was supported by the Ministry of Science and Technology, Government of the

## Abstract

### Background

Recently, the pumpkin by-products such as peels and seeds become more and more interesting for confectionary sector, because of the nutritional and bioactive composition. The study was conducted to develop cookies and noodles using different pumpkin by-products (such as peels and seeds) powders in different concentrations and to evaluate the changes in quality characteristics. The powders were prepared from pumpkin seeds and peels.

### Methodology

In this study, various concentrations of pumpkin peel powder (5%, 10%, and 20%) and seeds powder (5%, 10%, 15%, and 20%) were mixed with commercially available wheat flour to produce cookies and noodles. Furthermore, the nutritional, antioxidant, functional and sensory properties of the developed cookies and noodles were evaluated.

### Result

Pumpkin peels and seeds powder and their respective products enrich significantly in protein, fiber, beta-carotene, antioxidants such as total phenolic content (TPC), 2,2- diphenyl-1-picrylhydrazyl (DPPH scavenging activity), Ferric reducing ability (FRAP Ferric reducing power assay), ash content and hunter color values. However, pumpkin seed powder had higher protein content (33.25%) than other powders but pumpkin peel powder (16.83%±1.74) had better fiber content than seeds powder. The highest amount of fiber content was found in 10% pumpkin peel cookies. The analysis of the composition of pumpkin peels and seeds powder has been done in order to further promote their functionality in bakery products based, especially, on wheat flour.

### Conclusion

The best quality and overall acceptable product were obtained when samples were 10% pumpkin peel powder, 15% pumpkin seed powder, and, 10% pumpkin peel powder noodles.

Peoples' Republic of Bangladesh Special Allocation (Grant number ES 382; 2020-2021). The funders had no role in study design, data collection and analysis, decision to publish, or preparation of the manuscript.

## 1. Introduction

Pumpkin (*Cucurbita moschata*) is a fruit vegetable, not only grown in the Western Hemisphere but also in Tropical Asia in countries such as Bangladesh, India, Indonesia, Malaysia and Philippines [1]. The total world agricultural production has increased annually over the years from 111.986 million tons in 2018 to 136.27 million tons in 2020 while in Bangladesh pumpkin cultivation has an annual production of 340 thousand metric tons [1,2]. Pumpkins have a large range of uses as a potentially valuable food source for both humans and animals. The edible flesh (pulp) of the pumpkin fruit may be eaten as a vegetable or incorporated into food products such as curry, pies, bread, soup, or jams. The food industry utilizes pumpkin powders and flakes as thickeners, flavoring and coloring agents in soups and sauces [1,3]. However, food processing techniques create large quantities of waste in the form of peels (skin) and seeds. These waste streams can be utilized as an added value ingredient with the potential of being processed into food products in a similar way to other waste stream products and fiber-rich components [4–6]. Pumpkin peels and seeds are rich in vitamins, proteins, minerals and antioxidants (β-carotene and tocopherols) [7,8]. In addition to nutrient composition, it is composed of various biological active components such as polysaccharides, protein, peptide, sterols and para-amino benzoic acids [7,9]. These biological active components have shown to be wide range of medicinal properties such as anti-diabetic activity, antioxidant activity, anti-carcinogenic effect and anti- microbial effect [7,10].

With these beneficial effects of pumpkin peels and seeds, it can be incorporated into different value-added products such as cookies and snacks (noodles). In recent times, fruits and vegetables by-products have attracted more attention due to their health benefits including pumpkin pie, biscuits, bread, desserts, puddings, beverages, and soups. Pumpkins are celebrated in festivals and in flower and vegetable shows in many countries. Therefore, it has been focused on under-utilized indigenous crops waste, for example the pumpkin peels and seeds are useful in food industries for the formulation of value-added products. Several previous studies have demonstrated the strong antioxidant activity of pumpkin peels and seeds and their potential role in preventing hypertension and carcinogenic diseases [11–14]. Although there have been studies on formulations of cookies and noodles incorporating pumpkin peels and seeds, mixtures of wheat flour with various added ingredients are known to be rich in numerous active principles such as antioxidants, macro elements, microelements, and fiber, making them highly valuable raw materials for the confectionery industry [11].

In recent times, there has been increasing attention toward finding uses for by-products and waste generated from food processing. Utilizing these by-products would not only help maximize available resources but also introduce new food products to the market, while simultaneously addressing waste disposal issues.

Therefore, the objective of this study was to utilize pumpkin by-products and develop pumpkin peels and seeds powders mixed cookies and noodles and evaluation of its physical, chemical and sensory properties.

## 2. Materials and methods

### 2.1 Raw materials, chemicals and other ingredients

Fully ripe pumpkins (*Cucurbita moschata*) were sourced from a local market in Dhaka, Bangladesh chosen for uniformity, color, and lack of defects during July-august, 2023 harvest season. Pumpkin seeds, wheat flour (Local Brand: ACI Pure flour), oil, sugar, skimmed milk powder, butter, eggs, baking powder, salt, and margarine were obtained locally. Citric acid and packaging materials came from a local company. All items, except butter and eggs stored in a

refrigerator, were kept at room temperature. High-density polyethylene bags stored peel powder, seed powder, cookies, and noodle samples. All analytical grade chemicals were supplied by the food analytical laboratory and food processing laboratory, State University of Bangladesh, Dhaka.

## 2.2 Preparation of pumpkin peels and seeds powder

Ripe pumpkin was washed, sliced, and its peels collected. The peels were treated with 0.5% citric acid, dried at 65˚C for 8 hours. Purchased seeds were oven-dried at 80˚C for 22 hours [15,16]. Both dried peels and seeds (With husk) were blended by blender machine (Jaipan CM/L - 7360065), sieved by A.S.T.M. (E-11), Endecotts limited, London, England through a 50-mesh size, and packed in high-density polyethylene bags for analysis. The photo of the powders (Market flour, Seeds powder and peels powder) was added in the "Supporting Information files - Photo".

## 2.3 Preparation of cookies and noodles from pumpkin seeds and peels

**2.3.1 Powders formulation of cookies.**   Two types of cookies were formulated. Cookies were prepared from pumpkin seeds and peels powders. Pumpkin seeds powders and peels powders were substituted at 5%, 10%, 15%, 20% and 5%, 10%, 20%, respectively with marketed wheat flour which were represented in the Fig 1. Cookies prepared from market wheat flour serve as control. The quantities of the ingredients are listed in the Table 1. The photo of the cookies was added in the "Supporting Information files - Photo".

**2.3.2 Formulation of noodles.**   Noodles were also made from pumpkin seeds powder and pumpkin peels powders. Three types of noodles were formulated such as wheat four fortified with 10% seed powder, 10% peel powder and 10% seed powder mixed with 10% peel powder which were represented in the Fig 2. Noodles prepared from market wheat flour serve as control. The quantity of the noodle's preparation recipe is listed in the Table 2. The photo of the noodles was added in the "Supporting Information files - Photo".

## 2.4 Proximate composition

Moisture (oven drying method), protein (Kjeldahl method, where the N conversion factor is 6.25), fat (Soxhlet method), ash (Muffle furnace method) and crude fiber content of pumpkin seeds powder, peels powders, cookies and noodles were determined by official methods of Association of Official Analytical chemists (AOAC, 1984; 1998) [17] and the gluten content was determined according to the method reported by Taneya et al. (2014) [18].

Acid and alkali hydrolysis followed by ignition of the residue as described by the method AOAC (1984) was used to estimate the fiber content in pumpkin seeds & peels powders and cookies and noodles samples [17]. All the analyses were done in triplicates. Then the fiber content was determined by following equation

$$\% \ of \ Fibre - \frac{Initial \ weight - Final \ Weight}{Sample \ weight} \times 100$$

## 2.5 Determination of total polyphenol content

Determination of the total polyphenol content was done by Akter et al., (2010) method with some modifications [19]. 1 gram of samples was extracted using 20 ml of 25% ethanol for 15 minutes and subsequently filtered through Whatman no. 2 filter paper. Following this, a mixture of 1 ml of the sample, 1 ml of Folin reagent, and 5 ml of $Na_2CO_3$ was transferred to a volumetric flask and left at room temperature for 1 hour. Total phenols were calculated based on

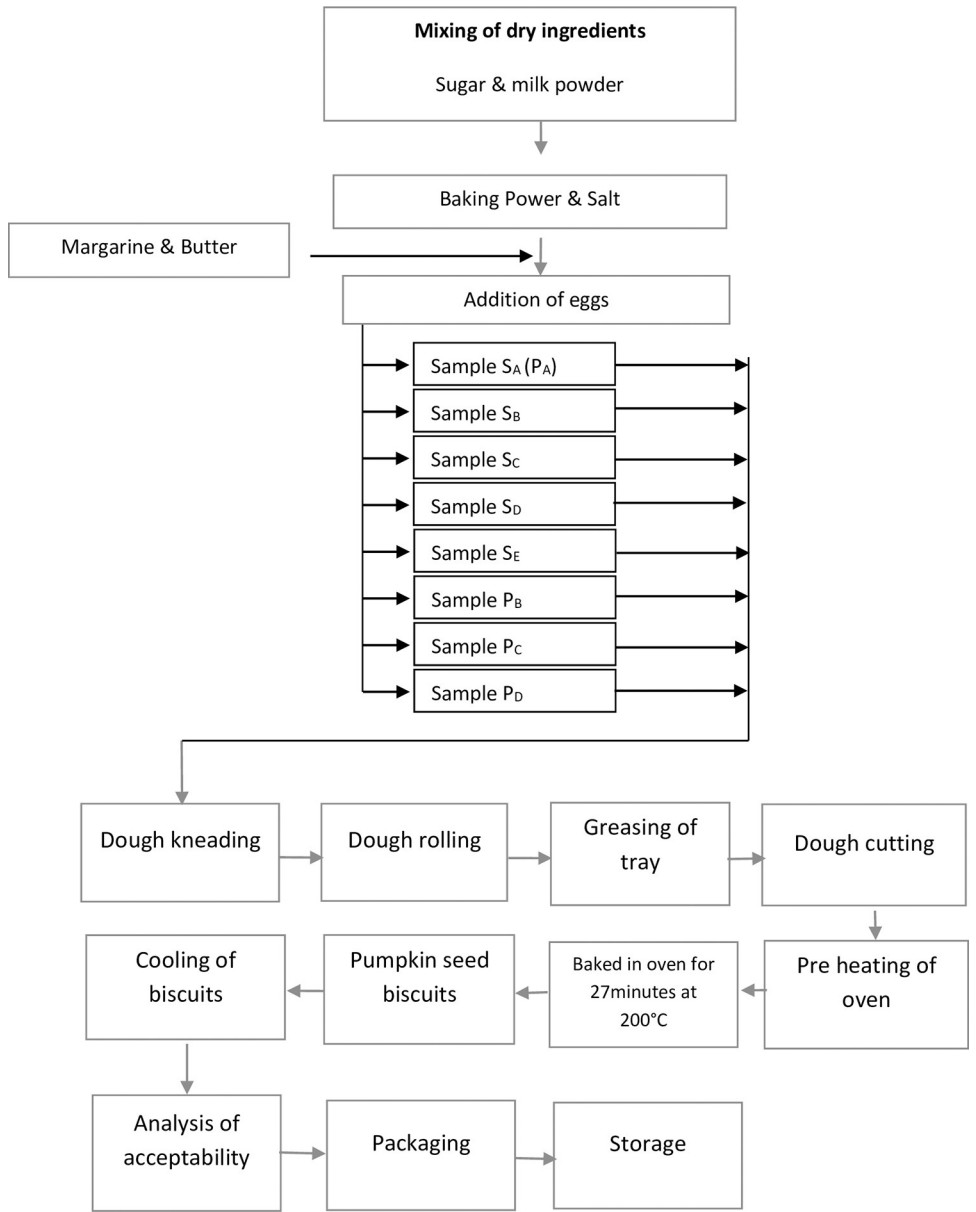

**Fig 1. Flow diagram of preparation of pumpkin peel & seed cookies samples.**

standard curves of gallic acid and expressed as mg/1 g utilizing spectrophotometer (T80 U/VIS, United Kingdom).

## 2.6 Determination of DPPH

The scavenging effects of the samples on 1,1-diphenyl-2-picrylhydrazyl (DPPH) radical were evaluated following a modified method based on the previous report by Akter et al. (2010) [19]. Each sample (1 milliliter) was combined with 5 milliliters of freshly prepared 0.1 mM DPPH methanolic solution. The resulting mixture was shaken and incubated in darkness for 50 minutes at room temperature. Absorbance was subsequently measured at 517 nm using a spectrophotometer (T80 U/VIS, United Kingdom). The DPPH radical scavenging activity was

**Table 1. The quantity of the different cookies.**

| Sample name | Ingredients | | | |
|---|---|---|---|---|
| | Pumpkin seeds (g) | Pumpkin peel (g) | Wheat (g) | Others |
| S$_A$ (P$_A$) (Control (100% market wheat flour, Cookie)) | 0 | 0 | 100 | Sugar 50g |
| S$_B$ (95% market wheat flour /5%pumpkin seed cookie) | 5 | 0 | 95 | Milk powder 5g Baking Power 2g |
| S$_C$ (90% market wheat flour /10%pumpkin seed cookie) | 10 | 0 | 90 | Salt 0.5g |
| S$_D$ (85% market wheat flour /15%pumpkin seed cookie) | 15 | 0 | 85 | Margarine 10g Butter 10g |
| S$_E$ (80%market wheat flour /20%pumpkin seed cookie) | 20 | 0 | 80 | Eggs 33g |
| P$_B$ (95%wheat/5%pumpkin peel cookie) | 0 | 5 | 95 | |
| P$_C$ (90%wheat/10%pumpkin peel cookie) | 0 | 10 | 90 | |
| P$_D$ (80%wheat/20%pumpkin peel cookie) | 0 | 20 | 80 | |

then calculated as follows:

$$Scavenging\ effect\ (\%) = [Abs_{control} - (Abs_{sample} - Abs_{sample\ backgroung})] \times 100/Abs_{control}$$

## 2.7 Determination of FRAP

The chelating capacity of PSE was assessed using a method outlined by Akter et al. (2010) [19]. Various concentrations of samples (0.1 mL) were combined with 0.1 mL of 2 mM FeCl$_2$·4H$_2$O and 0.2 mL of 5 mM ferrozine. Subsequently, 3.7 mL of MeOH was added to the mixture in a test tube and allowed to react for 10 minutes. The absorbance at 562 nm was then measured, and chelating activity was determined using the following equation:

$$Chelating\ activity\ (\%) = [Abs_{control} - (Abs_{sample} - Abs_{sample\ backgroung})] \times 100/Abs_{control}$$

## 2.8 Cooking properties

The cooking properties of noodles were evaluated by a method similar to that described by Mestres et al., 1988 [20]. Noodles weighing 1g were added to 150mL of boiling water and cooked for 1 minute to achieve optimal cooking. The sample was subsequently drained for 5 minutes and swiftly weighed (W$_1$, g). The cooked product was then dried in an oven at 130°C until a constant weight was achieved (W$_2$, g). The cooking water was centrifuged at 7500 rpm for 10 minutes, and the dry matter contents of the sediment and supernatant (W$_3$, g and W$_4$, g, respectively) were determined. These evaluations of cooking properties were all done in duplicate.

Cooking yield and losses were calculated with the following equations (DM = dry matter ratio of crude samples):

$$Cooking\ yield\ (\%) = \frac{(W_2) \times 100}{W_1}$$

$$Cooking\ loss\ (\%) = \frac{(DM - W_2) \times 100}{DM}$$

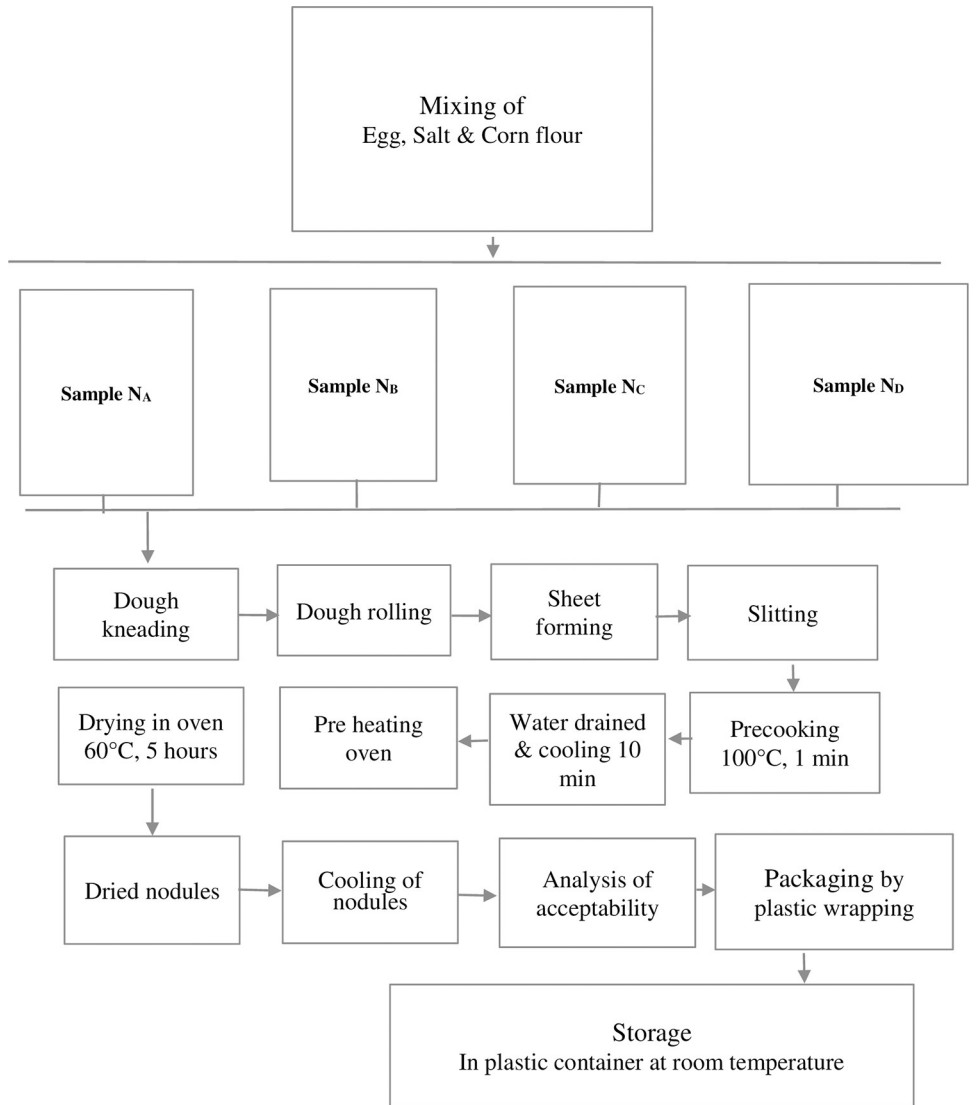

**Fig 2. Flow diagram of preparation of noodles samples.**

**Table 2. The quantity of the different noodles.**

| Sample name | Ingredients | | | |
|---|---|---|---|---|
| | Pumpkin seed (g) | Pumpkin peel (g) | Wheat (g) | Others |
| N$_A$ (Control (100% wheat flour noodles)) | 0 | 0 | 100 | Egg 33g Salt 0.5g Corn flour 2g |
| N$_B$ (90%wheat/10% pumpkin seeds noodles) | 10 | 0 | 90 | |
| N$_C$ (90%wheat/10% pumpkin peels noodles) | 0 | 10 | 90 | |
| N$_D$ (80%wheat/10%pumpkin seeds and 10% pumpkin peels noodles) | 10 | 10 | 80 | |

## 2.9 Sensory evaluation

A sensory analysis of cookies was conducted using a 9-point Hedonic scale by a panel of 10 members from the Department of Food Engineering and Technology at the State University of Bangladesh, Dhaka. The assessment covered attributes such as color, flavor, taste, texture, and overall acceptability. Similarly, sensory evaluations of both cookies and noodle samples were performed by a taste-testing panel, employing a hedonic rating test with ten panelists. They were tasked with evaluating various sensory parameters and assigning scores on a nine-point hedonic scale, where 9 represented 'Like extremely,' 8 for 'Like very much,' and so forth. Preference differences were subsequently evaluated through statistical analysis of the data for variance.

## 2.10 Statistical analysis

The results were expressed as the mean, standard deviation, and coefficient of variation of each species for each parameter was determined. All evaluations of samples were all done in triplicate. Data were statistically analyzed (R statistical software 3.4.1) by one-way analysis of variance (ANOVA). Statistical analysis for the sensory analysis, including variance and Duncan's Multiple Range Test (DMRT), was performed using the Statistical Analysis System (SAS, 1985) procedures. Mean comparisons were performed using Duncan's multiple range tests for significant effect at $P < 0.05$.

## 3. Result and discussion

### 3.1 Proximate analysis

**3.1.1 Proximate analysis of wheat flour, pumpkin seeds and peels powder.** The three kinds of powders were analyzed for proximate analyses which are wheat flour which collected from local market, pumpkin seed powder and pumpkin peel powder. The results showed in the Table 3 where results showed significantly higher amounts of moisture, ash, protein, fat, fiber and beta-carotene contents in pumpkin seeds powder and peels powder as compared to marketed wheat flour (control). Similar results were observed for pumpkin seeds powder and peels powder by Hussain et al., 2022; who reported that moisture, ash, protein, fat and fiber content of pumpkin seeds powder was 2.75%, 5.20%, 34.19%, 44.01% and 4.12%, respectively whereas pumpkin peels powder was 10.31%, 6.04%, 12.30%, 5.77% and 13.91%, respectively [21]. Sometimes the difference in protein and fat value in pumpkin seeds and peels were observed. This might be due to the variation in environment during harvesting such as application of fertilizer (nitrogen content), growing conditions and time and also location of growing areas, oxidation of the fat because most of the fat content in seeds and peels is unsaturated and undergo oxidation easily by atmospheric oxygen. Atmospheric oxidation normally product of alkyl peroxyls of hydrocarbons. Normally when the concentration is low, they can react

**Table 3. Proximate compositions of powder samples.**

| Powder Samples | Moisture Content (%) | Ash Content (%) | Protein Content (%) | Fat Content (%) | Fiber Content (%) | Beta- carotene (µg/g) | Gluten (%) |
|---|---|---|---|---|---|---|---|
| Wheat flour | 10.60±0.87[a] | 0.57±0.02[b] | 9.62 ±0.01[b] | 1.08±0.03[b] | 0.51±0.09[b] | 14.56±0.34[b] | 14.45±0.76[a] |
| Pumpkin seed powder | 3.21±0.16[b] | 5.18±0.03[a] | 33.25±0.01[a] | 43.96±0.33[a] | 3.91±0.48[b] | 39.46±0.36[b] | 0 |
| Pumpkin peel powder | 10.21±0.13[a] | 4.40±0.03[a] | 15.09±0.30[ab] | 2.94±0.01[b] | 16.83±1.74[a] | 123.55±0.33[a] | 0 |

All values are express as mean ± SD.

Mean followed by different superscript letters in each column are significantly different (p<0.05).

**Table 4. Proximate compositions of pumpkin seeds cookies.**

| Cookie Samples | Moisture content (%) | Ash content (%) | Protein content (%) | Fat content (%) | Fiber content (%) | Beta- carotene (ug/g) |
|---|---|---|---|---|---|---|
| $S_A$ | 6.68 ±0.19[bc] | 0.91±0.01[c] | 8.09 ±1.54[c] | 11.62±0.56[d] | 0.32±0.15[c] | 4.07±0.36[c] |
| $S_B$ | 5.91±0.31[c] | 1.02±0.03[c] | 9.84±0.31[b] | 12.03±0.02[cd] | 0.62±0.19[bc] | 5.18±0.34[b] |
| $S_C$ | 7.79±0.56[ab] | 1.24±0.01[ab] | 9.84 ±0.31[b] | 12.89±0.01[bc] | 0.92±0.06[b] | 5.27±0.39[ab] |
| $S_D$ | 7.96± 0.06[a] | 1.15±0.02[b] | 11.15±0.31[a] | 13.95±0.70[a] | 1.61±0.31[a] | 5.77±0.74[ab] |
| $S_E$ | 7.74±0.54[ab] | 1.36±0.03[a] | 12.03±0.31[a] | 13.78±0.86[ab] | 1.52±0.30[a] | 6.36±1.08[a] |

Values are means of triplicate analysis ± Standard deviation. $S_A$ = Control (100% market wheat flour, Cookie), $S_B$ = 95% market wheat flour /5%pumpkin seed cookie, $S_C$ = 90% market wheat flour /10%pumpkin seed cookie, $S_D$ = 85% market wheat flour /15%pumpkin seed cookie, $S_E$ = 80%market wheat flour /20%pumpkin seed cookie).

All values are express as mean±SD.

Mean followed by different superscript letters in each column are significantly different (p<0.05).

with themselves which is called self-reaction of alkyl peroxyl radicals. If the concentration is high, in the unsaturated condition the oxygen molecules are added in the subsequent stage, and the self-reactions can be lead to aldehydes and alcohols [22]. The gluten content of the wheat flour which collected from local renowned company was 14.47% while pumpkin seeds and peels powder was no gluten, which also we found that the gluten value of Bangladesh around 11% to 15% [23].

**3.1.2 Proximate analysis of pumpkin seed cookies.** The proximate compositions of cookies from pumpkin seeds were analyzed for moisture, ash, protein, fat, fiber and beta-carotene which presented in Table 4. Results demonstrated that all pumpkin seed cookies had higher amount of protein, fat, fiber and beta-carotene content as compared to control (100% wheat flour cookie).

Protein content of different cookies varied from 8.09 to 12.03/100 g that was similar to found by Alshehry (2020), reported protein value of in pumpkin seed cookies 12.62 to 13.75% on the variation of the percentage of the pumpkin seeds powder using to develop, Syam et al. (2019) who reported protein value in pumpkin seed cookie 11.20/100g and also by Dhiman et al. (2018) who reported protein value in cookies as 8.47 to 12.56/100g [24–26]. Sample $S_E$ had highest protein value (12.03/100g) compared with other recipes. On the contrary, sample $S_A$ which is backed by market wheat flour had the lowest value of protein (8.09/100 g) than other samples. The variations in protein value increase because pumpkin seed contain more protein than market wheat flour.

Fat values were appreciably different among cookies and highest was found in sample $S_D$ (13.95/100g), where lowest content (11.62/100g) were studied in sample $S_A$. This observation was comparable to that found by Alshehry (2020) who reported fat value in pumpkin seed cookies 7.13/100g [24]. The variation in fat value in different cookies may be due to oxidation of the fat because most of the fat content in cookies is unsaturated and undergo oxidation easily by atmospheric oxygen. The ash content of different cookies varied between 0.91/100 to 1.36/100g. This result was consistent to be found by Alshehry (2020) and Syam et al. (2019) who reported ash content in cookies as 1.62/100g and 1.65/100g respectively [24,25]. Lowest ash value was found in sample $S_A$ which is backed from market wheat flour (0.91/100g) variety while highest value (1.36/100g) was studied in sample $S_E$ which backed from 85% market wheat flour /15%pumpkin seed cookie.

Fiber values were appreciably different among cookies and highest was found in sample $S_D$ (1.61/100g), where lowest content (0.32/100g) were studied in sample $S_A$. This observation was comparable to that found by Apostol et al. (2020), Alshehry (2020) and Syam et al. (2019) who

reported fat value in pumpkin seed cookies 5.74/100g, 6.24/100g and 1.64/100g respectively [11,24,25]. Low molecular weight components such as minerals, vitamins and sugars might be lost during washing treatment, leading to a relative increase in the dietary fibre content (Wenberg et al., 2006) [27]. The Beta-carotene content of different cookies varied between 4.07 ug/g to 6.36 ug/g. This result was consistent to be found by Dhiman et al. (2018) who reported Beta-carotene content in cookies as 3.22 to 3.78 mg/100g [26]. Lowest Beta-carotene content was found in sample $S_A$ which is backed from market wheat flour (4.07ug/g) variety while highest value (6.36ug/g) was studied in sample $S_E$ which backed from 80% market wheat flour 20% pumpkin seed cookie.

**3.1.3 Proximate compositions of pumpkin peel cookies.** The composition of cookies from pumpkin peels was analyzed for moisture, ash, protein, fat, fiber and beta-carotene. The results presented in Table 5 where sample $P_D$ contained the highest amount of moisture and sample $P_A$ had the lowest level of moisture. These values were higher to those reported by Mishra and Sharma (2019) who found that moisture content in cookies is 6.59/100g and also by Saleh and Ali (2020) found that moisture content in pumpkin peel cookie was 5.62/100g to 7.09/100g [28,29]. Once more, Hussain et al. (2023) observed comparable moisture content levels in pumpkin flash biscuits, ranging from 6.69% to 7.04%, with variations in flash content ranging from 5% to 15% [30]. Typically, elevated levels of moisture have been linked to the limited duration baked goods can be stored, as they promote the entry of microbes, resulting in their decay [31].

Protein content of different pumpkin peel cookies varied from 8.09 to 9.04/100g that was similar to found by Saleh and Ali (2020) who reported protein value in pumpkin peel cookie 7.73/100g to 9.42/100g and higher found by Mishra and Sharma (2019) who reported protein value in cookies as 0.08/100g [28,29]. However, Hussain et al. (2023) reported a lower protein value compared to other studies, with values ranging slightly below at 6.62% to 6.22% [30]. Sample $P_D$ had highest protein value (9.40/100g) compared with other recipes. On the contrary, sample $P_A$ which is backed by market wheat flour had the lowest value of protein (8.09/100 g) than other samples. The variations in protein value increase because pumpkin peel contains more protein than market wheat flour.

Fat values were appreciably different among cookies and highest was found in sample $P_C$ (11/100g), where lowest content (10.84/100g) were studied in sample $P_A$. This observation was comparable to that found by Mishra and Sharma (2019) who reported fat value in pumpkin peel cookies 21.75/100g and also found by Saleh and Ali (2020) who reported fat value in pumpkin peel cookie 14.82/100g to15.14/100g [28,29]. Hussain et al. (2023) found that pumpkin flash biscuits exhibited a higher fat content. Their studies reported fat contents ranging from 31.13% to 31.46% [30]. The variation in fat value in different cookies may be due to

**Table 5. Proximate compositions of pumpkin peel cookies.**

| Cookie Samples | Moisture Content (%) | Ash Content (%) | Protein Content (%) | Fat Content (%) | Fiber Content (%) | Beta- carotene (ug/g) |
|---|---|---|---|---|---|---|
| $P_A$ | 6.68 ±0.19[c] | 0.91±0.01[c] | 8.09 ±1.54[a] | 11.62±0.56[a] | 0.32±0.15[c] | 4.07±0.36[c] |
| $P_B$ | 7.17±0.17[bc] | 1.04±0.01[b] | 8.75±0.61[a] | 10.84±0.16[b] | 2.07±0.19[bc] | 12.40±0.36[b] |
| $P_C$ | 7.63±0.52[b] | 1.10±0.02[b] | 8.96 ±0.31[a] | 11.00±0.21[ab] | 5.08±0.06[a] | 25.52±0.19[a] |
| $P_D$ | 9.02± 0.43[a] | 1.34±0.01[a] | 9.40±0.30[a] | 10.95±0.13[ab] | 3.08±0.31[ab] | 29.38±0.37[a] |

Values are means of triplicate analysis ± Standard deviation. $P_A$ = Control (100% wheat, Cookie), $P_B$ = 95%wheat/5%pumpkin peel cookie, $P_C$ = 90%wheat/10% pumpkin peel cookie, $P_D$ = 80%wheat/20%pumpkin peel cookie).

All values are express as mean ± SD.

Mean followed by different superscript letters in each column are significantly different($p<0.05$).

oxidation of the fat because most of the fat content in cookies is unsaturated and undergo oxidation easily by atmospheric oxygen. The ash content of different cookies varied between 0.91/100 to 1.34/100g. This result was consistent to be found by Saleh and Ali (2020) who reported ash content in cookies as 1.30/100g to 2.62/100g [29]. In their studies, Hussain et al. (2023) observed similar ash content, ranging from 1.09% to 1.32%, across different flash content variations from 5% to 15% [30]. Lowest ash value was found in sample $P_A$ which is backed from market wheat flour (0.91/100 g) variety while highest value (1.34/100 g) was studied in sample $P_D$ which backed from 85% market wheat flour /15%pumpkin peel cookie.

Fiber values were appreciably different among cookies and highest was found in sample $P_C$ (5.08/100g), where lowest content (0.32/100g) were studied in sample $P_A$. This observation was comparable to that found by Saleh and Ali (2020) who reported fiber value in pumpkin peel cookies 1.18/100g to3.17/100g and higher then found by Mishra and Sharma (2019) who reported fiber value in pumpkin peel cookies 0.16/100g [28,29].

The Beta-carotene content of different cookies varied between 4.07ug/g to 29.38ug/g. This result was higher to be found by Saleh and Ali (2020) who reported Beta-carotene content in cookies as 3.08 to 11.54 mg/100g [29]. Lowest Beta-carotene content was found in sample A which is backed from market wheat flour (4.07ug/g) variety while highest value (39.38ug/g) was studied in sample $P_D$ which backed from 80% market wheat flour /20% pumpkin peel cookie. The variation in beta-carotene value increase because pumpkin peel contains more beta-carotene than market wheat flour.

**3.1.4 Proximate compositions of noodles samples.** The composition of noodles from pumpkin seeds and peels powder was analyzed for moisture content, protein, ash and fat content. The results obtained were presented in Table 6. Sample had lower moisture content as compared to all other peels and seeds fortified noodles. Although these values had no significant differences. The protein content of noodles sample ranged from 4.81 to 5.47%. There were no significant differences in protein content among all noodles sample. All noodles' samples had fat content ranged from 0.59 to 4.96%. Sample $N_D$ (80% wheat/ 10% pumpkin seeds and 10% pumpkin peels noodles) had highest fat content which is slightly higher than control. The ash content of noodles sample ranged from 1.60 to 2.27%. Sample $N_D$ (80% wheat/ 10% pumpkin seeds and 10% pumpkin peels noodles) sample had highest ash content. There was significantly different in ash content was observed for sample $N_D$. This might be due to the higher amount of mineral content present in the seeds and peels samples. The fiber content of noodles sample ranged from 1.49 to 8.88%. Control (wheat flour noodles) sample had lower fiber content as compared to all other peels and seeds fortified noodles.

**3.1.5 β-Carotene determination for noodles.** Almeida et al. (2011) reported that, foods which are rich in antioxidants play an essential role in the prevention of carcinogenic diseases

**Table 6. Proximate compositions of noodles samples.**

| Noodles Samples | Moisture Content (%) | Ash Content (%) | Protein Content (%) | Fat Content (%) | Fiber Content (%) | Beta-carotene (ug/g) |
|---|---|---|---|---|---|---|
| $N_A$ | 8.94±0.62[a] | 1.60±0.026[b] | 5.47 ±0.31[a] | 4.30±0.47[a] | 1.49 ±0.23[a] | 0.324 ±0.003[a] |
| $N_B$ | 9.67±0.97[a] | 1.85±0.20[b] | 5.03±0.30[a] | 0.59 ±0.01[a] | 6.36 ±1.58[a] | 0.315 ±0.004[a] |
| $N_C$ | 9.44±0.85[a] | 1.72±0.034[b] | 4.81±0.02[a] | 4.96 ±0.44[a] | 8.88±1.13[a] | 0.222±0.004[b] |
| $N_D$ | 9.23±0.42[a] | 2.27±0.140[a] | 5.25 ±0.03[a] | 0.59± 0.07[a] | 8.82 ± 0.19[a] | 0.318 ± 0.002[a] |

Values are means of triplicate analysis ± Standard deviation. $N_A$ = Control (100% wheat flour noodles), $N_B$ = 90%wheat/10% pumpkin seeds noodles, $N_C$ = 90%wheat/10% pumpkin peels noodles, $N_D$ = 80%wheat/10%pumpkin seeds and 10% pumpkin peels noodles.

All values are express as mean±SD.

Mean followed by different superscript letters in each column are significantly different($p<0.05$).

[32]. The antioxidant capacities of samples such as pumpkin peels and seeds may vary depending on their contents of vitamin C, vitamin E, carotenoids, and particularly β-carotene [33], and lycopene [34] as well as other pigments such as flavonoids and other polyphenols [35]. The results for β-carotene content of the noodles are listed on Table 6. Carotenoids are tetraterpenoids found throughout not only the flowering plant kingdom but also in vegetables and fruits as a pigment mostly responsible for various color such as red, orange or yellow color in peels, seeds, flash leaves and fruits which are important as vitamin A precursors. As they are found widely in all kinds' plants and plant parts, it is not surprising that a large number of carotenoids have been reported in the pumpkin seeds and peels powder [35]. Noodles from pumpkin seeds powder showed the highest (P < 0.05) content of β-carotene except control. Noodles manufactured from peels powder and mixture of peels and seeds have also shown to be an excellent source of β-carotene. Similar levels of β-carotene have been previously reported for pumpkin flesh powder [36,37]. However, these pumpkin peels and seeds powder should not be considered as a rich source of carotenoids, where values as high as 161mg/100g d.b. for wine palm (*Mauritia vinifera*) one of the most important vitamin A precursors has been represented [38,39].

## 3.2 Cooking properties (cooking yield)

Generally speaking, high cooking yield gains and low cooking loss would be desirable characteristics for high quality of any kinds of noodles. Flour noodles with added different types of pumpkin powder such as peels powder, seeds powder and mixture of peels and seeds powders had lower cooking yield gains than the flour noodle samples with no added pumpkin (Table 7).

It was found that control sample ($N_A$) contained the highest amount of cooking yield and 10% seeds powder noodles sample ($N_B$) had the lowest level of cooking yield while 10% pumpkin seeds powder has lowest cooking loss and the mix powder has highest cooking loss. Including controls all noodles' samples cooking loss is considerable higher than Apostol et al. (2020) and Mestres et al. (1988) [14]. But all noodles' samples cooking yield is much higher [37].

The parameters of cooked noodles varied considerably for gumminess and hardness. However, springiness, cohesiveness, and chewiness were not different among the samples, indicating the pumpkin powder did not affect such parameters [37,40].

## 3.3 Hunter color values

Hunter color values were represented by L*, a*, b*, and ΔE where L* is a measure of lightness, a* redness, b* yellowness and ΔE color differences. The Hunter color parameters L*, a*, b*,

**Table 7. Cooking properties of noodles.**

| Sample | Noodles cooking Yield (%) | Noodles cooking Loss (%) |
|--------|---------------------------|--------------------------|
| $N_A$ | 275.254±0.10[a] | 3.723±0.010[b] |
| $N_B$ | 256.368±0.10[b] | 3.421 ±0.010[b] |
| $N_C$ | 250.885±0.10[b] | 4.217±0.010[ab] |
| $N_D$ | 260.089±0.01[b] | 5.007±0.002[a] |

Values are means of triplicate analysis ± Standard deviation. $N_A$ = Control (100% wheat flour noodles), $N_B$ = 90% wheat/10% pumpkin seeds noodles, $N_C$ = 90%wheat/10% pumpkin peels noodles, $N_D$ = 80%wheat/10%pumpkin seeds and 10% pumpkin peels noodles.

All values are express as mean±SD.

Mean followed by different superscript letters in each column are significantly different(p<0.05).

**Table 8. Hunter Color Values (L\*, a\*, b\*, ΔE) of pumpkin seeds and peels powder samples.**

| Powder Samples | L* | a* | b* | ΔE |
|---|---|---|---|---|
| Wheat Flour | 80.53±2.87[a] | 1.82±0.19[a] | 10.57±0.55[b] | 81.24±2.92[a] |
| Pumpkin seed powder | 52.79±1.24[b] | 3.57±0.74[a] | 19.46±1.35[a] | 56.39±1.77[b] |
| Pumpkin peel powder | 65.09±4.74[ab] | 1.34±0.26[a] | 18.72±4.27[a] | 67.78±5.73[ab] |

All values are express as mean±SD.

Mean followed by different superscript letters in each column are significantly different(p<0.05).

and ΔE have been widely used to evaluate color changes during dehydration of fruits and vegetable products. There are three types of powdered samples such as wheat flour which collected from local market served as control, Pumpkin seed powder and Pumpkin peel powder. The results of Hunter Color Values (L\*, a\*, b\*, ΔE) of pumpkin seeds and peels powder are represented in Table 8, pumpkin seed cookies are shown in Table 9 and pumpkin peels cookies are shown in Table 10.

Results indicates lightly decrease in L\*, b\* and ΔE value while increase in a\* value of all cookies samples prepared from both seeds and peels powders with the level of increasing sample concentrations. The increase of hunter a\* values (redness) with increase of sample concentration for all cookie samples might be due to the influence of beta-carotene concentration. The flour colors and cookies color can best be described by the change in total color difference, ΔE values, where locally collected wheat flour has a significance difference with both pumpkin seeds and peels powder. Pumpkin seeds cookies $S_A$, $S_E$ and pumpkin peels cookies $P_A$, $P_B$ and $P_c$ shows almost similar results The lower ΔE values may be due to the loss of phenolics and oxidation during drying, boiling treatment and oven temperature [41].

## 3.4 Antioxidant properties

**3.4.1 Total phenolic compound and antioxidant properties (DPPH scavenging activity and FRAP Ferric reducing power assay) of powder samples.** The total phenolic compound, DPPH and FRAP antioxidant properties of market wheat flour, pumpkin seed powder and pumpkin peel powder are shown in Table 11. The highest total phenolic compound (268.38mg/100g) was found in pumpkin seed powder sample. It is commonly known as seeds are an excellent source of nutritional compositions and other bioactive compound specially phenols but pumpkin peels also very excellent source of antioxidant which also supported by

**Table 9. Hunter color values (L\*, a\*, b\*, ΔE) of pumpkin seeds cookies.**

| Cookie Samples | L* | a* | b* | ΔE |
|---|---|---|---|---|
| $S_A$ | 50.94±1.85[ab] | 16.20±0.80[a] | 28.46±0.45[a] | 60.56±1.98[ab] |
| $S_B$ | 49.14±1.90[ab] | 13.51±0.96[b] | 26.88±0.41[ab] | 57.621±2.04[ab] |
| $S_C$ | 44.04±2.93[b] | 14.40±1.45[ab] | 23.40±2.43[b] | 51.91±3.98[b] |
| $S_D$ | 43.02±0.41[b] | 14.46±0.48[ab] | 24.50±0.82[ab] | 51.58±.87[b] |
| $S_E$ | 54.07±1.67[a] | 13.95±1.18[ab] | 27.60±0.13[ab] | 62.29±1.77[a] |

Values are means of triplicate analysis ± Standard deviation. $S_A$ = Control (100% market wheat flour, Cookie), $S_B$ = 95% market wheat flour /5%pumpkin seed cookie, $S_C$ = 90% market wheat flour /10%pumpkin seed cookie, $S_D$ = 85% market wheat flour /15%pumpkin seed cookie, $S_E$ = 80%market wheat flour /20%pumpkin seed cookie).

All values are express as mean ± SD.

Mean followed by different superscript letters in each column are significantly different (p<0.05).

**Table 10. Hunter color values (L\*, a\*, b\*, ΔE) of pumpkin peels cookies.**

| Cookie Samples | L* | a* | b* | ΔE |
|---|---|---|---|---|
| $P_A$ | 50.94±1.85[a] | 16.20±0.80[a] | 28.46±0.45[a] | 60.56±1.98[a] |
| $P_B$ | 55.84±1.60[a] | 10.67±0.19[b] | 28.97±0.24[a] | 63.81±1.54[a] |
| $P_C$ | 51.19±1.13[a] | 9.70±0.72[b] | 29.48±0.86[a] | 59.86±1.51[a] |
| $P_D$ | 41.50±2.34[b] | 13.25±0.61[ab] | 24.08±2.27[b] | 49.78±3.21[b] |

Values are means of triplicate analysis ± Standard deviation. $P_A$ = Control (100% wheat, Cookie), $P_B$ = 95%wheat/5% pumpkin peel cookie, $P_C$ = 90%wheat/10%pumpkin peel cookie, $P_D$ = 80%wheat/20%pumpkin peel cookie).

All values are express as mean ± SD.

Mean followed by different superscript letters in each column are significantly different (p<0.05).

the Hagos et al. 2023 where he reported the phenolic compound are almost four times higher the pumpkin seeds in pumpkin peels [24,42]. The variety of growth circumstances, plant parts, processing, analytical methods and environment, could all be considered in explaining the considerable variances in the phenolic contents [24,42].

**3.4.2 Total phenolic compound and antioxidant properties (DPPH scavenging activity and FRAP Ferric reducing power assay) of pumpkin seeds cookies.** Table 12 shows the total phenolic compound and antioxidant properties (DPPH and FRAP) of pumpkin seeds cookies. All pumpkin seeds cookies comparatively showed higher total phenolic compound as compared to control (market wheat flour). This is because of higher total phnolic compound present in pumpkin seeds samples. As a result, pumpkin seeds powders fortification in wheat flour cookies can be excellent substitute of wheat flour to enrich nutrients. It was observed that pumpkin seeds cookies had higher DPPH scavenging activity as compared to control (market wheat flour). On the other hand, FRAP antioxidant properties of pumpkin seeds cookies samples ranged from 1157.25 to 2079.75 mmole/100g. It was also evaluated that pumpkin seeds cookies had higher FRAP antioxidant properties as compared to control (market wheat flour). The higher values of DPPH and FRAP antioxidant activity of pumpkin seeds fortified cookies samples might be due to the presence of higher phenolic compounds in pumpkin seeds.

**3.4.3 Total phenolic compound and antioxidant properties (DPPH scavenging activity and FRAP Ferric reducing power assay) of pumpkin peels cookies.** Table 13 represents the total phenolic compound and antioxidant properties (DPPH and FRAP) of pumpkin peels cookies. All pumpkin peels cookies showed higher total phenolic compound as compared to control (market wheat flour). This is because of higher total phenolic compound present in pumpkin peels samples. DPPH scavenging activity of pumpkin peels cookies samples ranged from 111.50 to 199.29 mmole/100g. The highest amount of DPPH properties (199.29 m mole/100g) was observed for the cookie sample containing 20% peel powder (sample $S_D$). Ferric

**Table 11. Total phenolic compound and antioxidant properties (DPPH and FRAP) of market wheat flour, pumpkin seed powder and pumpkin peel powder sample.**

| Powder Samples | Total Phenolic Compound (mg/100g) | DPPH scavenging activity (mmole/100g) | FRAP Ferric reducing power assay (mmole/100g) |
|---|---|---|---|
| Wheat Flour | 83.62±2.85[b] | 122.23±6.86[c] | 279.75±37.12[c] |
| Pumpkin seed powder | 268.38±1.02[a] | 160.17±14.55[b] | 688.50±10.60[b] |
| Pumpkin peel powder | 87.82±1.03[b] | 217.52±27.76[a] | 1374.75±15.90[a] |

All values are express as mean ± SD.

Mean followed by different superscript letters in each column are significantly different (p<0.05).

**Table 12. Total phenolic compound and antioxidant properties (DPPH and FRAP) of pumpkin seeds cookies.**

| Cookie Samples | Total Phenolic Compound (mg/100g) | DPPH scavenging activity (m mole/100g) | FRAP Ferric reducing power assay (m mole/100g) |
|---|---|---|---|
| $S_A$ | $24.51^a \pm 0.57$ | $162.52^{ab} \pm 12.47$ | $1187.25^b \pm 58.33$ |
| $S_B$ | $82.57^a \pm 0.01$ | $117.97^b \pm 18.09$ | $1157.25^b \pm 26.51$ |
| $S_C$ | $76.37^a \pm 0.22$ | $196.35^{ab} \pm 39.09$ | $1269.75^b \pm 47.72$ |
| $S_D$ | $71.75^a \pm 2.74$ | $286.64^a \pm 22.46$ | $2079.75^a \pm 79.54$ |
| $S_E$ | $60.28^a \pm 1.14$ | $179.00^{ab} \pm 5.40$ | $1873.50^a \pm 265.1$ |

Values are means of triplicate analysis ± Standard deviation. $S_A$=Control (100% market wheat flour, Cookie), $S_B$=95% market wheat flour /5%pumpkin seed cookie, $S_C$=90% market wheat flour /10%pumpkin seed cookie, $S_D$=85% market wheat flour /15%pumpkin seed cookie, $S_E$=80%market wheat flour /20%pumpkin seed cookie).

All values are express as mean ± SD.

Mean followed by different superscript letters in each column are significantly different ($p<0.05$).

reducing power (FRAP) assay was evaluated for pumpkin peels cookies which was ranged from1029.75to1187.25 mmole/100g. It was evaluated that the market wheat flour (control) had the highest (1187.25 mmole/100g) FRAP value.

**3.4.4 Total phenolic compound and antioxidant properties (DPPH) of noodles samples.** Table 14 displayed total phenolic compound of all noodles sample ranged from 9.55 to 34.96 mg/100g. Sample $N_B$ had the highest total phenolic compound (34.96mg/100g). On the other hand, DPPH scavenging activity of all noodles sample ranged from 14.03 to 27.08 mmole/100g. Sample $N_C$ had the highest DPPH scavenging activity (27.08mmole/100g). All noodles' samples had higher DPPH scavenging activity than control. It indicated that wheat flour can be fortified with pumpkin seeds and peels powders to enrich nutritional properties specially antioxidant activities.

## 3.5 Hydration properties of powder samples

Hydration properties such as water absorption index, water solubility index and swelling capacity of market wheat flour, pumpkin seed powder and pumpkin peel powder are shown in Table 15. The values of all powder samples ranged from 187.40 to 764.60 g/100g, 5.60 to 17.60 ml/100g and 5.93 to 21.37ml/100g for water absorption index, water solubility index and swelling capacity, respectively. Hussain et al. (2023) reported 60.31/100g and 60.98/100g water absorption index of the pumpkin peels powder and pumpkin seeds powder [43]. The data showed pumpkin peels powder and seeds powder had hydration properties which is close to the market wheat flour (control) sample. It indicates that these two-powder ingredients (seeds

**Table 13. Total phenolic compound and antioxidant properties (DPPH and FRAP) of pumpkin peels cookies.**

| Cookie Samples | Total Phenolic Compound (mg/100g) | DPPH scavenging activity (mmole/100g) | FRAP Ferric reducing power assay (mmole/100g) |
|---|---|---|---|
| $P_A$ | $24.51^b \pm 0.57$ | $162.52^a \pm 12.47$ | $1187.25^b \pm 58.33$ |
| $P_B$ | $80.47^a \pm 2.51$ | $138.55^a \pm 13.51$ | $1048.50^a \pm 63.63$ |
| $P_C$ | $46.48^{ab} \pm 1.02$ | $111.50^a \pm 7.27$ | $1063.50^a \pm 30.75$ |
| $P_D$ | $60.45^{ab} \pm 5.02$ | $199.294^a \pm 25.37$ | $1029.75^a \pm 5.30$ |

Values are means of triplicate analysis ± Standard deviation. $P_A$=Control (100% wheat, Cookie), $P_B$=95%wheat/5%pumpkin peel cookie, $P_C$=90%wheat/10%pumpkin peel cookie, $P_D$=80%wheat/20%pumpkin peel cookie).

All values are express as mean ± SD.

Mean followed by different superscript letters in each column are significantly different ($p<0.05$).

**Table 14. Total phenolic compound and antioxidant properties (DPPH) of noodles samples.**

| Noodles Samples | Total Phenolic Compound (mg/100g) | DPPH scavenging activity (mmole/100g) |
|---|---|---|
| $N_A$ | 18.86±0.49[ab] | 14.03±2.54[b] |
| $N_B$ | 34.96±0.27[a] | 16.28±1.79[b] |
| $N_C$ | 21.82±1.09[ab] | 27.08±1.27[a] |
| $N_D$ | 9.55±1.09[b] | 18.75±1.42[ab] |

Values are means of triplicate analysis ± Standard deviation. $N_A$=Control (100% wheat flour noodles), $N_B$=90% wheat/10% pumpkin seeds noodles, $N_C$=90%wheat/10% pumpkin peels noodles, $N_D$=80%wheat/10%pumpkin seeds and 10% pumpkin peels noodles.

All values are express as mean ± SD.

Mean followed by different superscript letters in each column are significantly different (p<0.05).

powder and peels powder) can be partially replaced by wheat flour for making less gluten cookie. It is impossible completely reduced the gluten by the seeds and peels powder but we can enrich the noodles nutrition value by partially adding the waste materials of pumpkin.

## 3.6 Sensory evaluation

Ten judges' evaluated cookies and noodles made with different ratios of pumpkin powder based on texture, aroma, color, taste, and overall acceptability. Tables 16–18 present mean scores for each criterion. Two-way ANOVA for color preference indicated significant differences (p<0.05) among the samples. However, DMRT showed no significant taste differences in pumpkin peel cookies. Among peel cookies, sample $P_C$ scored highest in color, aroma, texture, and overall acceptability. In pumpkin seed cookies, sample $S_D$ with 15% powder achieved the highest scores. For noodles, sample $P_C$ with 10% pumpkin peel powder outperformed others, securing the highest scores in color, texture, aroma, taste, and overall acceptability. The findings suggest optimal ratios for enhancing sensory attributes in pumpkin-enriched cookies and noodles.

The sensory characteristics of cookies containing varying levels of pumpkin seeds are detailed in Table 16. The results of sensory evaluation reveal significant variations among different cookie products. Notably, Sample $P_C$ (90% wheat/10% pumpkin peel cookie) achieved the highest ratings in color, aroma, taste, texture, and overall acceptability, scoring 8, 7.33, 7.67, 8, and 8.33, respectively, as illustrated in Table 17. Similarly, Sample $S_D$ (85% market wheat flour/15% pumpkin seed cookie) demonstrated superior attributes, scoring highest in color, aroma, taste, texture, and overall acceptability, with ratings of 7.50, 7.67, 8, 7.67, and 8.33, respectively, as shown in Table 18. Additionally, in Table 16, $N_C$ (90% wheat/10% pumpkin peels noodles) emerged with the highest score among others.

**Table 15. Hydration properties (water absorption index, water solubility index and swelling capacity of powder samples).**

| Cookie Samples | Water Absorption Index (g/100g) | Water Solubility Index (ml/100g) | Swelling Capacity (ml/100g) |
|---|---|---|---|
| Wheat Flour | 764.60±20.64[a] | 17.60±1.13[a] | 21.37±1.66[a] |
| Pumpkin seed powder | 240.40±6.70[b] | 11.20±0.56[b] | 9.53±0.71[b] |
| Pumpkin peel powder | 187.40±0.20[b] | 5.60±0.01[c] | 5.93±0.01[c] |

All values are express as mean ± SD.

Mean followed by different superscript letters in each column are significantly different (p<0.05).

**Table 16. Mean score for texture, aroma, color, taste and overall acceptability of pumpkin peel cookies.**

| Cookie's type | Color | Aroma | Taste | Texture | Overall acceptability |
|---|---|---|---|---|---|
| P$_A$ | 7.67±0.99[a] | 7.00±0.85[ab] | 7.00±0.86[a] | 7.33±0.98[a] | 7.33±1.31[b] |
| P$_B$ | 7.67±0.49[a] | 6.58±0.51[bc] | 6.91±0.80[a] | 7.33±1.30[a] | 7.66±0.98[ab] |
| P$_C$ | 8.00±0.86[a] | 7.33±0.49[a] | 7.67±0.99[a] | 8.00±0.85[a] | 8.330.49±[a] |
| P$_D$ | 6.67±0.99[b] | 6.33±0.50[c] | 7.17±0.84[a] | 6.33±0.50[b] | 6.00±0.01[c] |

Values are means of triplicate analysis ± Standard deviation. P$_A$=Control (100% wheat, Cookie), P$_B$=95%wheat/5%pumpkin peel cookie, P$_C$=90%wheat/10%pumpkin peel cookie, P$_D$=80%wheat/20%pumpkin peel cookie).

All values are express as mean ± SD.

Mean followed by different superscript letters in each column are significantly different (p<0.05).

**Table 17. Mean score for texture, aroma, color, taste and overall acceptability of pumpkin seed cookies.**

| Cookie's type | Color | Aroma | Taste | Texture | Overall acceptability |
|---|---|---|---|---|---|
| S$_A$ | 6.42±0.51[b] | 7.33±1.30[ab] | 7.00±0.85[b] | 7.33±0.98[a] | 7.33±1.30[b] |
| S$_B$ | 6.58±0.50[b] | 6.67±0.49[b] | 6.33±0.98[b] | 6.33±0.49[b] | 7.33±1.30[b] |
| S$_C$ | 6.33±0.77[b] | 6.67±0.98[b] | 6.67±0.49[b] | 7.00±0.85[ab] | 7.00±0.85[b] |
| S$_D$ | 7.50±0.67[a] | 7.67±0.98[a] | 8.00±0.85[a] | 7.67±0.78[a] | 8.33±0.49[a] |
| S$_E$ | 6.58±0.51[b] | 7.00±0.96[ab] | 6.67±0.50[b] | 7.00±0.85[ab] | 7.33±1.32[b] |

Values are means of triplicate analysis ± Standard deviation. S$_A$=Control (100% market wheat flour, Cookie), S$_B$=95% market wheat flour /5%pumpkin seed cookie, S$_C$=90% market wheat flour /10%pumpkin seed cookie, S$_D$=85% market wheat flour /15%pumpkin seed cookie, S$_E$=80%market wheat flour /20%pumpkin seed cookie).

All values are express as mean ± SD.

Mean followed by different superscript letters in each column are significantly different (p<0.05).

**Table 18. Mean score for texture, aroma, color, taste and overall acceptability of pumpkin seed and peel noodles.**

| Noodles type | Color | Aroma | Taste | Texture | Overall acceptability |
|---|---|---|---|---|---|
| N$_A$ | 7.67±0.98[a] | 7.25±1.21[a] | 7.00±0.84[a] | 7.33±0.98[a] | 7.33±1.31[bc] |
| N$_B$ | 7.67±0.49[a] | 7.33±0.50[a] | 7.00±0.95[a] | 7.33±1.30[a] | 7.67±0.99[ab] |
| N$_C$ | 8.00±0.85[a] | 7.67±0.65[a] | 7.67±0.99[a] | 8.00±0.85[a] | 8.33±0.50[a] |
| N$_D$ | 6.67±0.98[b] | 6.33±0.49[b] | 6.00±0.01[b] | 6.33±0.49[b] | 6.58±0.58[c] |

Values are means of triplicate analysis ± Standard deviation. N$_A$=Control (100% wheat flour noodles), N$_B$=90%wheat/10% pumpkin seeds noodles, N$_C$=90%wheat/10% pumpkin peels noodles, N$_D$=80%wheat/10%pumpkin seeds and 10% pumpkin peels noodles.

All values are express as mean ± SD.

Mean followed by different superscript letters in each column are significantly different (p<0.05).

The enhancement in texture upon increasing pumpkin seeds in cookies may be attributed to their high protein content, as observed by Atuonwu and Akobundu (2010) [44]. The color score ranged from 6.33 in control cookies to 7.50 in cookies containing 20% pumpkin seeds, reflecting the desirable effect of pumpkin seeds' high protein content. Furthermore, the gradual decrease in the color score of cookies may be attributed to sugar caramelization and Maillard reactions between sugars and amino acids, consistent with findings by Alshehry et al. (2020) indicating an improvement in color score with higher supplementation levels [24,44].

Cookies exhibited significant variations in flavor, with the highest score observed in those containing 15% pumpkin seeds and the lowest in control cookies. This increase in flavor was also noted by Atuonwu and Akobundu (2010) with higher levels of pumpkin seeds. Moreover,

the highest taste score was found in products containing 15% pumpkin seeds, emphasizing the importance of pumpkin seed inclusion for developing taste characteristics in cookies (Murkovic, 2004) [44,45].

Studies by Alshehry (2020) and Kanwal et al. (2015) demonstrated successful incorporation of pumpkin seed flour as a partial replacement for wheat flour, resulting in cookies with enhanced nutritional value and wholesomeness without compromising overall acceptability [24,46].

## 4. Conclusions

Enriching confectionery products with pumpkin seeds and peels enhances their nutritional value. This process also ensures that the products have satisfactory technological and sensory characteristics. The effects of seeds and peels on the physicochemical and nutritional characteristics of different confectionary products such as cookies and noodles were investigated. Pumpkin seed and peel powders could be used not only enrich nutritious value but also to enhance the quality of food products antioxidant properties such as total phenolic compounds where all pumpkin seeds (95% market wheat flour /5%pumpkin seed cookie has 82.57 mg/100g) and peels sample (95% market wheat flour /5%pumpkin peel cookie has 80.47 mg/100g) showed the enrichment of the confectionary products. The increase in fibre and ash content in the cookies and noodles showed qualitative and quantitative advantages when compared to the market flour. The sensory acceptability rate reached 78%, which refers to "Good" in the hedonic rating scale, which is considered good acceptability. Although pumpkin production is not widespread throughout the country and the economic conditions in Bangladesh do not support this type of industry, the results demonstrate the production viability of this alternative food due to its very low cost and ease of preparation. More importantly, it can be used in various meals, thereby improving nutritional intake.

## Author Contributions

**Conceptualization:** Md. Asaduzzaman.

**Data curation:** Md. Asaduzzaman.

**Formal analysis:** Mahbubur Rahman.

**Investigation:** Md. Asaduzzaman.

**Methodology:** Md. Asaduzzaman, Mahbubur Rahman.

**Resources:** Md. Asaduzzaman.

**Supervision:** Mst. Sorifa Akter.

**Visualization:** Md. Asaduzzaman.

**Writing – original draft:** Md. Asaduzzaman.

**Writing – review & editing:** Mst. Sorifa Akter.

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
