## [Decision Letter · Decision Letter 0]

16 Feb 2024

PONE-D-23-44103Enhancing Nutritional Value and Quality of Cookies through Pumpkin Peel and Seed Powder Fortification: A Wheat Flour Replacement ApproachPLOS ONE

Dear Dr. Asaduzzaman,

Thank you for submitting your manuscript to PLOS ONE. After careful consideration, we feel that it has merit but does not fully meet PLOS ONE’s publication criteria as it currently stands. Therefore, we invite you to submit a revised version of the manuscript that addresses the points raised during the review process.

**Please revise the manuscript carefully and provide detailed responses for each comment.**

We look forward to receiving your revised manuscript.

Kind regards,

Hongju He

Academic Editor

PLOS ONE

A clean copy of the edited manuscript (uploaded as the new *manuscript* file).

“Special Projects from Ministry of Science and Technology

Environmental Science: 382.”

Reviewers' comments:

Reviewer's Responses to Questions

**Comments to the Author**

1. Is the manuscript technically sound, and do the data support the conclusions?

Reviewer #1: No

Reviewer #2: Yes

Reviewer #3: Yes

2. Has the statistical analysis been performed appropriately and rigorously? 

Reviewer #1: No

Reviewer #2: No

Reviewer #3: Yes

3. Have the authors made all data underlying the findings in their manuscript fully available?

Reviewer #1: Yes

Reviewer #2: Yes

Reviewer #3: Yes

4. Is the manuscript presented in an intelligible fashion and written in standard English?

Reviewer #1: No

Reviewer #2: Yes

Reviewer #3: Yes

5. Review Comments to the Author

Reviewer #1: The manuscript entitled “Enhancing Nutritional Value and Quality of Cookies through Pumpkin Peel and Seed Powder Fortification: A Wheat Flour Replacement Approach” is interesting. However, it lacks novelty and is poorly written. It is not suitable for publication in its current form. I have the following comments to improve this manuscript:

Title:

Is it really going to be “A Wheat Flour Replacement Approach”? It's tough to say with such a low amount of seeds and peel powder. Besides, pumpkin seeds are very small amounts, considering the massive amount of wheat flour produced worldwide. I suggest deleting this portion: “A Wheat Flour Replacement Approach.”

Abstract:

-“Significant differences (p<0.05) in protein, fibre, beta……..”---This statement is not clear. For this sentence, we do not know if the amounts are higher or lower or if they have a positive or negative impact on product quality. Please make it clear.

As you mentioned % of peel, seed, etc., in the results and conclusion of the abstract, you should add this to the methodology, too.

You have also mentioned Overall acceptability in the results of the abstract but did not mention it in the methods of the abstract.

Introduction:

-Use updated data. You used data from 2005, and now it’s 2024!

-The introduction section is very short; it lacks literature study and references. The novelty of the work has not been stated properly.

Material and Methods:

-Section 2.2: Were the seeds husked before milling?

-Using both Figures 1 and 2 is unnecessary. You can merge them and mention the seed/peel portion in different samples. Otherwise, you may discard both of the figures and illustrate them in your writing.

-Section 2.4 and 2.5: Where does sweet potato powder come here?

-Section 2.4: If the Protein determination process is through the Kjeidhal method, please mention the N conversion factor.

-Section 2.5-2.9: You must have to add all the details about the methodology. These methods are not reproducible and not suitable for publication.

-Section 2.10: The sensory analysis should be vividly described. Also, the number of panellists needs to be higher. Add whether they were trained/semi-trained, how samples were presented to them, etc. The statistical analysis portion should be moved to section 2.11.

-Section 2.11: Mention how many replicates for each observation were used.

Results and Discussion:

-Table 01: Check 3.9.1±0.48b?

-In methodology, you mentioned Sample 1, 2, 3, etc…in Results and Discussion, you are saying Sample A, B, C, etc…

-If every result of your study is comparable to those of Syam et al. (2019), then what is the justification of your study?

-Where is the Carbohydrate content in your results???

-Section 3.2: “The antioxidant capacities of leaves????????? may vary………..”? You should be careful while writing a manuscript for publication.

-Besides, most of the information in this section is irrelevant. It would help if you thought about your samples and raw ingredients.

-Section 3.3: Add the units of cooking loss and cooking yield. From the standard deviation, it seems the indications of significant differences are not correct.

-Table 12: The results of sample D are not homogenous with the other results. Besides, where are the FRAP results of Noodles?

-Section 3.6: You should mention the significance of Hydration properties. The discussion of this section is also inadequate. Like other parameters, you didn’t measure the hydration properties of your samples.

-Section 3.7: Previously, you mentioned 10 panellists!

-Table 14, 15, 16: You need to add the standard deviations in the results. Besides, you may show them by spider graph for better visibility.

-Besides, this section lacks discussion.

-Minerals content: You did not mention this in the methodology. If you would like to keep it, you should add it before sensory evaluation. Obviously, you should add mineral contents to all samples and ingredients as well. You should improve the discussion section also.

Conclusions:

-The first sentence is not clear. Make it simple.

References:

Most of the references used in the manuscripts are too old. You should use more updated references.

Overall comments:

There are some typos and grammatical errors. Authors are advised to revise the entire manuscript to make it publishable.

Reviewer #2: The article deals with a topical issue. The replacement of the proportion of wheat in bakery products is still topical and leads to the production of healthier and greener products. Pumpkin pulp and seeds are an impact product, where their further use is clearly related to the rational use of waste products in production plants.

I recommend the article for publication after incorporating the comments.

P11 2.1. the variety and degree of ripeness should be mentioned. The city of Dhaka should also be mentioned with the country. The wheat flour used should be characterised more precisely, gliadin, protein, starch content or technological property should be mentioned.

Figures 1-3 have different labelling of the samples, there are mixed samples1-4, formula1-4 and in the results are discus samples A-B. This should be standardised. This figure also shows the abbreviation gm, this should be explained in the note, or if it is g, it should be changed.

P13 Figure 2 the weight of the ingredients should be stated separately "Water & eggs 33g".

P15 2.4 and 25 why is it written "sweet potato peel"?

P16 For used equipment the manufacturer and country of origin should be stated. The source of the chemical used should also be mentioned.

P17 Table 1 shows no statistical differences between the samples, but the mean is different and the SD is low. The author should explain this.

P18 biscuits33.05/100g - missing space

P18 2nd column (C) and P19 3C - the author should explain how fat oxidation can change fat content or delete this sentence.

P18 "Dietary fibre values were significantly different between the cookies and the highest was found in sample D (1.61/100g), while the lowest content (0.32/100g) was investigated in sample A. This observation was comparable to that of Syam et al. (2019) who reported a fat value of 1.64/100g in pumpkin seed cookies [17]." - Correctly described ingredients

P21 Chapter 3.2 should be part of 3.1 as in previous biscuits.

P22 Table 05 sample labelling should be identical in manuscript,

P23 Table 06 use the same orientation in the cell, for Delta E there are no significant differences but the values are different and the SD is low, this should be explained.

P24 1C I did not agree that seeds are a better source of polyphenols. This should be confirmed by literature or discussed in more detail. In my opinion there are interferences with proteins in seeds.

P24 Table 09 were there no statistical differences?

P24 3.5.2 and Table 10 should explain why Sample C has the lowest TPC and the highest antioxidant activity, this is not consistent with the results in Table 09.

P25 F RAP value - correct to FRAP value.

P25 The sentence "showed higher total phenolic compound compared to control" is misleading, the statistical differences were not found for all samples.

P26 3.5.4 and Table 12 do not describe FRAP, it should be stated why it is not used or added.

P27 The fact that the water absorption index, water solubility index and swelling capacity are high does not mean that they are capable of forming a 3D network. The sentence "It indicates that these two powdered ingredients (seed powder and hull powder) can be replaced by wheat flour for making less gluten biscuits" should be more clearly described or deleted.

P28 Tables 14 and 15: standardise the labelling of the samples, 6.00c- use the superscript.

P29 Effect of mineral compounds on nutrition should be discussed with literature.

P30 Conclusion, antioxidant activity should be mentioned, rheological properties were not instrumentally analysed so should not be mentioned in the conclusion.

Reviewer #3: Rewrite the abstract, introduction and conclusions.

Need to add photographs of the products and raw material used in the materials.

The study is unique .....such studies needs to promoted.

This work not only prompts healthy foods but also creates economic impacts in food industry.

6. PLOS authors have the option to publish the peer review history of their article (what does this mean?). If published, this will include your full peer review and any attached files.

Reviewer #1: No

Reviewer #2: No

Reviewer #3: **Yes: **Dr M.S.ASKI

---

## [Author Response · Author response to Decision Letter 0]

20 Mar 2024

Detailed response to reviewers

Manuscript Reference: PONE-D- 23-44103

Manuscript Title: Enhancing Nutritional Value and Quality of Cookies through Pumpkin Peel and Seed Powder Fortification.

Author(s): Md. Asaduzzaman1,2, Mahbubur Rahman1, Mst. Sorifa Akter3*

1) Department of Food Engineering and Technology, State University of Bangladesh, Dhaka, Bangladesh.

2) Department of Food Processing and Preservation, Hajee Mohammad Danesh Science and Technology University, Dinajpur, Bangladesh

3) Department of nutrition and food Engineering, Daffodil International University, Dhaka, Bangladesh.

*Corresponding Author

Dear Editor and Reviewers,

Thank you very much for your constructive and useful suggestions. Those comments were valuable and very helpful for revising and improving our paper, aswell as for increasing the significance of our research. The responses to editor andreviewer’s comments/suggestions of our manuscript have been answered carefullypoint-by-point as follows. Some minor mistakes within the manuscript have alsobeen corrected.

Note: All the line numbers in our responses refer to the revised manuscript. Thered color in the manuscript indicates the changes made during revision.

Response to reviewers

Reviewer #1:

Thank you for your positive perspective on the manuscript.

The comments of the reviewer

Title:

Is it really going to be “A Wheat Flour Replacement Approach”? It's tough to say with such a low amount of seeds and peel powder. Besides, pumpkin seeds are very small amounts, considering the massive amount of wheat flour produced worldwide. I suggest deleting this portion: “A Wheat Flour Replacement Approach.”

Comments of the answer:

According to the suggestion of the reviewer, “A Wheat Flour Replacement Approach” is deleting from the title.

Abstract:

-“Significant differences (p<0.05) in protein, fibre, beta……..”---This statement is not clear. For this sentence, we do not know if the amounts are higher or lower or if they have a positive or negative impact on product quality. Please make it clear.

As you mentioned % of peel, seed, etc., in the results and conclusion of the abstract, you should add this to the methodology, too.

You have also mentioned Overall acceptability in the results of the abstract but did not mention it in the methods of the abstract.

Comments of the answer:

The abstract is rewritten according to the suggestion of the reviewer 01. The methodology is added in the abstract.

Introduction:

-Use updated data. You used data from 2005, and now it’s 2024!

-The introduction section is very short; it lacks literature study and references. The novelty of the work has not been stated properly.

Comments of the answer:

The updated data added in the introduction section and data are updated. There more literature and references added in the introduction.

Material and Methods:

-Section 2.2: Were the seeds husked before milling?

Comments of the answer: we did not husked the seeds before milling, we add the husked to in the cookies and noodles development.

-Using both Figures 1 and 2 is unnecessary. You can merge them and mention the seed/peel portion in different samples. Otherwise, you may discard both of the figures and illustrate them in your writing.

Comments of the answer: While it is true that both figures share many commonalities, combining them would result in overly complicated design and complexity in understanding. Hence, we have chosen to maintain them as separate figures. However, should you suggest amalgamating them into one figure; we will provide detailed representation of the leveling process in the methodology section.

The samples in Figures 1 to 3 originally had disparate labeling, which has now been standardized uniformly under the same notation.

SA = Control (100% market wheat flour, Cookie)

SB = 95% market wheat flour /5%pumpkin seed cookie

SC = 90% market wheat flour /10%pumpkin seed cookie

SD = 85% market wheat flour /15%pumpkin seed cookie

SE = 80%market wheat flour /20%pumpkin seed cookie

PA = Control (100% market wheat flour, Cookie)

PB = 95%wheat/5%pumpkin peel cookie

PC = 90%wheat/10%pumpkin peel cookie

PD = 80%wheat/20%pumpkin peel cookie

NA = Control (100% wheat flour noodles)

NB = 90%wheat/10% pumpkin seeds noodles

NC = 90%wheat/10% pumpkin peels noodles

ND = 80%wheat/10%pumpkin seeds and 10% pumpkin peels noodles

-Section 2.4 and 2.5: Where does sweet potato powder come here?

Comments of the answer: Mistakenly we wrote “sweet potato peel” which corrected to pumpkin seeds powder, peels powders, cookies and noodles in the section 2.4 and 2.5.

-Section 2.4: If the Protein determination process is through the Kjeidhal method, please mention the N conversion factor.

Comments of the answer: In the section 2.4 – the N conversion factor is added. 

-Section 2.5-2.9: You must have to add all the details about the methodology. These methods are not reproducible and not suitable for publication.

Comments of the answer: according to your suggestion all section 2.5 to 2.8 are described. One section is added to the previous section, so that, one section is minimized.

-Section 2.10: The sensory analysis should be vividly described. Also, the number of panellists needs to be higher. Add whether they were trained/semi-trained, how samples were presented to them, etc. The statistical analysis portion should be moved to section 2.11.

Comments of the answer: The sensory analysis methodology has been updated in accordance with the reviewer's suggestions. The statistical analysis section has been relocated to Section 2.10. All panelists are faculty members of the Department of Food Engineering and Technology, and prior to the sensory analysis, they underwent training for the tests. I regret the limited number of panelists; however, currently, our department lacks qualified sensory panelists. We are actively working to increase the number of panelists and are organizing training sessions to recruit and train new members.

-Section 2.11: Mention how many replicates for each observation were used.

Comments of the answer: All evaluations of samples were all done in triplicate.

Results and Discussion:

-Table 01: Check 3.9.1±0.48b?

Comments of the answer: the data was corrected. It will be 3.91±0.48b

-In methodology, you mentioned Sample 1, 2, 3, etc…in Results and Discussion, you are saying Sample A, B, C, etc…

Comments of the answer: The sample indications were wrong which was corrected and described according to the data in the methodology section.

-If every result of your study is comparable to those of Syam et al. (2019), then what is the justification of your study?

Comments of the answer: Along with syam et al. (2019), more literature added in the section 3.1.2. such as Alshehry (2020) and Apostol et al. (2020)

-Where is the Carbohydrate content in your results???

Comments of the answer: There are no carbohydrate discussions in this study, so the carbohydrate portion was removed from methodology.

-Section 3.2: “The antioxidant capacities of leaves????????? may vary………..”? You should be careful while writing a manuscript for publication.

Comments of the answer: thank you the suggestion, we will be more careful to write the manuscript for publishing in this renowned journal.

-Section 3.3: Add the units of cooking loss and cooking yield. From the standard deviation, it seems the indications of significant differences are not correct.

Comments of the answer: We add more literature in this section to differentiate differences.

-Table 12: The results of sample D are not homogenous with the other results. Besides, where are the FRAP results of Noodles?

Comments of the answer: The description is corrected and due to the some difficulties in the laboratory the FRAP test are not include for the noodles sample.

-Section 3.6: You should mention the significance of Hydration properties. The discussion of this section is also inadequate. Like other parameters, you didn’t measure the hydration properties of your samples.

Comments of the answer: in this section firstly we discussed the importance of the hydration properties and then we add more discussion and literatures.

-Section 3.7: Previously, you mentioned 10 panellists!

Comments of the answer: Mistakenly we count our self as a panelist which was corrected.

-Table 14, 15, 16: You need to add the standard deviations in the results. 

-Besides, this section lacks discussion.

Comments of the answer: The standard deviation is added in table 14 to 16. Description and literature is also added.

-Minerals content: You did not mention this in the methodology. If you would like to keep it, you should add it before sensory evaluation. Obviously, you should add mineral contents to all samples and ingredients as well. You should improve the discussion section also.

Comments of the answer: The section on mineral content has been removed because no other confectionery items besides noodles and samples were examined for minerals, which is not relevant to this study.

Conclusions:

-The first sentence is not clear. Make it simple.

Comments of the answer: The conclusion sentence has been simplified for brevity. The antioxidant activity is discussed in the corresponding section.

References:

Most of the references used in the manuscripts are too old. You should use more updated references.

Comments of the answer: In the references section, updated refinances are added which cited in the study.

Reviewer #2:

P11 2.1.the variety and degree of ripeness should be mentioned. The city of Dhaka should also be mentioned with the country. The wheat flour used should be characterised more precisely, gliadin, protein, starch content or technological property should be mentioned.

Comments of the answer:

In Section 2.1, in compliance with the reviewer's suggestion, Dhaka is now accompanied by the country name in the author's affiliation and the materials and methods section. Additionally, the gluten value was included in the manuscript following the powder test as per the reviewer's recommendation. Furthermore, the scientific name of the pumpkin variety and the degree of ripeness are detailed in the materials and methods section.

Figures 1-3 have different labelling of the samples, there are mixed samples1-4, formula1-4 and in the results are discus samples A-B. This should be standardised. This figure also shows the abbreviation gm, this should be explained in the note, or if it is g, it should be changed.

Comments of the answer:

Figure 1 to 3 had different labeling of the samples which now converted in to same notation and standardised uniformly where 

SA = Control (100% market wheat flour, Cookie)

SB = 95% market wheat flour /5%pumpkin seed cookie

SC = 90% market wheat flour /10%pumpkin seed cookie

SD = 85% market wheat flour /15%pumpkin seed cookie

SE = 80%market wheat flour /20%pumpkin seed cookie

PA = Control (100% market wheat flour, Cookie)

PB = 95%wheat/5%pumpkin peel cookie

PC = 90%wheat/10%pumpkin peel cookie

PD = 80%wheat/20%pumpkin peel cookie

NA = Control (100% wheat flour noodles)

NB = 90%wheat/10% pumpkin seeds noodles

NC = 90%wheat/10% pumpkin peels noodles

ND = 80%wheat/10%pumpkin seeds and 10% pumpkin peels noodles

P13 Figure 2 the weight of the ingredients should be stated separately "Water & eggs 33g".

Comments of the answer: In the figure 02, the weight of the ingredients mistakenly wrote water and egg 33g. It should be only egg 33 g which is corrected in the figure.

P15 2.4 and 25 why is it written "sweet potato peel"?

Comments of the answer: Mistakenly we wrote “sweet potato peel” which corrected to pumpkin seeds powder, peels powders, cookies and noodles in the section 2.4 and 2.5.

P16 For used equipment the manufacturer and country of origin should be stated. The source of the chemical used should also be mentioned.

Comments of the answer: The equipment manufacturer and origin are added in the 2.2 section including country. The source of the chemical used in the study also included in the section 2.1.

P17 Table 1 shows no statistical differences between the samples, but the mean is different and the SD is low. The author should explain this.

Comments of the answer: The statistical differences between the samples are corrected and the explanation is added in the section 3.1.1.

P18 biscuits33.05/100g - missing space

Comments of the answer: the space is added between biscuits 33.05/100g in the section 3.1.2.

P18 2nd column (C) and P19 3C - the author should explain how fat oxidation can change fat content or delete this sentence.

Comments of the answer: The explanation of fat oxidation is added in the section 3.1.1. This also repeated in the section 3.1.2.

P18 "Dietary fibre values were significantly different between the cookies and the highest was found in sample D (1.61/100g), while the lowest content (0.32/100g) was investigated in sample A. This observation was comparable to that of Syam et al. (2019) who reported a fat value of 1.64/100g in pumpkin seed cookies [17]." - Correctly described ingredients

Comments of the answer: The sample indications were wrong which was corrected and described according to the data.

P21 Chapter 3.2 should be part of 3.1 as in previous biscuits.

Comments of the answer: chapter 3.2 is changed to 3.1.5

P22 Table 05 sample labelling should be identical in manuscript,

Comments of the answer: Table 05 sample labeling is corrected and wrote according to the formulation figure and data.

P23 Table 06 use the same orientation in the cell, for Delta E there are no significant differences but the values are different and the SD is low, this should be explained.

Comments of the answer: in the table 06, the leveling data is corrected and the explanations are added in the section 3.3.4. More literature is added describe in the section.

P24 1C I did not agree that seeds are a better source of polyphenols. This should be confirmed by literature or discussed in more detail. In my opinion there are interferences with proteins in seeds.

Comments of the answer: the pumpkin peels are more better source of the total phenolic compounds then pumpkin seeds which was discussed in the section 3.4.1

P24 Table 09 were there no statistical differences?

Comments of the answer: the statistical significance added in the table 09 and discussed in the section 3.4.1.

P24 3.5.2 and Table 10 should explain why Sample C has the lowest TPC and the highest antioxidant activity, this is not consistent with the results in Table 09.

Comments of the answer: the value of the sample Sc was incorrectly imputed.

P25 F RAP value - correct to FRAP value.

Comments of the answer: the “F RAP value” corrected to “FRAP value”.

P25 The sentence "showed higher total phenolic compound compared to control" is misleading, the statistical differences were not found for all samples.

Comments of the answer: the sentence was corrected.

P26 3.5.4 and Table 12 do not describe FRAP, it should be stated why it is not used or added.

Comments of the answer: Due to the some difficulties in the laboratory the FRAP test are not include for the noodles sample.

P27 The fact that the water absorption index, water solubility index and swelling capacity are high does not mean that they are capable of forming a 3D network. The sentence "It indicates that these two powdered ingredients (seed powder and hull powder) can be replaced by wheat flour for making less gluten biscuits" should be more clearly described or deleted.

Comments of the answer: in the section 3.5, the sentence was changed and more literature and description was added.

P28 Tables 14 and 15: standardise the labelling of the samples, 6.00c- use the superscript.

Comments of the answer: the tables labeling and the superscript are corrected.

P29 Effect of mineral compounds on nutrition should be discussed with literature.

Comments of the answer: the tables labeling and the superscript are corrected.

P30 Conclusion, antioxidant activity should be mentioned, rheological properties were not instrumentally analysed so should not be mentioned in the conclusion.

Comments of the answer: The sentence of the conclusion describe briefly and make it simple. The ant

---

## [Decision Letter · Decision Letter 1]

24 Apr 2024

PONE-D-23-44103R1Enhancing Nutritional Value and Quality of Cookies through Pumpkin Peel and Seed Powder FortificationPLOS ONE

Dear Dr. Asaduzzaman,

Thank you for submitting your manuscript to PLOS ONE. After careful consideration, we feel that it has merit but does not fully meet PLOS ONE’s publication criteria as it currently stands. Therefore, we invite you to submit a revised version of the manuscript that addresses the points raised during the review process. ==============================

**Please revise and provide detailed responses.** 

We look forward to receiving your revised manuscript.

Kind regards,

Hongju He

Academic Editor

PLOS ONE

Journal Requirements:

Additional Editor Comments:

The manuscript titled “Enhancing Nutritional Value and Quality of Cookies through Pumpkin Peel and Seed Powder Fortification” presented by Md. Asaduzzaman discussed the novel approach to incorporate pumpkin seed powder as raw material for cookies and noodle and to provide a sustainable way of producing the food. Overall manuscript written well but I feel the manuscript has serious flows in presentation and formatting.

Author should carefully check the manuscript.

Reference- Author has marked reference with both numbered and author et al., . Author should read the Journal instruction for reference.

There are many small small paragraph that needs to be merged along with content for understanding of the readers.

Discussion stills needs improvement.

A detailed reviewer report mark in pdf file, kindly refer to pdf page.

General comment:

• At several places there is spacing issue, authors are suggested to correct it

• Minor English and grammar mistakes are present that needs to be thoroughly checked to improve the quality of the manuscript. Additionally, there are few sentences that are too large for the readers with many comma ( , ), that needs author attention.

Reviewers' comments:

Reviewer's Responses to Questions

**Comments to the Author**

1. If the authors have adequately addressed your comments raised in a previous round of review and you feel that this manuscript is now acceptable for publication, you may indicate that here to bypass the “Comments to the Author” section, enter your conflict of interest statement in the “Confidential to Editor” section, and submit your "Accept" recommendation.

Reviewer #2: All comments have been addressed

2. Is the manuscript technically sound, and do the data support the conclusions?

Reviewer #2: Yes

3. Has the statistical analysis been performed appropriately and rigorously? 

Reviewer #2: Yes

4. Have the authors made all data underlying the findings in their manuscript fully available?

Reviewer #2: Yes

5. Is the manuscript presented in an intelligible fashion and written in standard English?

Reviewer #2: Yes

6. Review Comments to the Author

Reviewer #2: All recommendations have been accepted. I have recommended the article for publication. Only one comment: in chapter 3.4.3, first column, m

mole/100g is in bold and should be in normal font.

7. PLOS authors have the option to publish the peer review history of their article (what does this mean?). If published, this will include your full peer review and any attached files.

Reviewer #2: **Yes: **Matej Pospiech

---

## [Author Response · Author response to Decision Letter 1]

8 May 2024

RESPONSE TO EDITOR

01. Comments:

The manuscript titled “Enhancing Nutritional Value and Quality of Cookies through Pumpkin Peel and Seed Powder Fortification” presented by Md. Asaduzzaman discussed the novel approach to incorporate pumpkin seed powder as raw material for cookies and noodle and to provide a sustainable way of producing the food. Overall manuscript written well but I feel the manuscript has serious flows in presentation and formatting.

Answer of the Comments: Thank you for your positive perspective on the manuscript.

02. Comments:

Author should carefully check the manuscript.

Reference- Author has marked reference with both numbered and author et al., . Author should read the Journal instruction for reference. There are many small paragraph that needs to be merged along with content for understanding of the readers. Discussion stills needs improvement.

Answer of the Comments: All reference is corrected including marked in the introduction section and written according to the journal instruction. According to the suggestion the small paragraphs are merged. We try our best to improve the discussion section by adding more references and secondary data.

03. Comments:

A detailed reviewer report mark in pdf file, kindly refer to pdf page.

Answer of the Comments: The suggestions are corrected according to reviewer and in needed next time, all answers and remarks will be written on the basis of PDF page.

04. General comments:

At several places there is spacing issue, authors are suggested to correct it

Answer of the Comments: The spacing issues are corrected including keywords which were happened due to the version change of the Microsoft word.

05. Comments: Minor English and grammar mistakes are present that needs to be thoroughly checked to improve the quality of the manuscript. Additionally, there are few sentences that are too large for the readers with many comma ( , ), that needs author attention.

Answer of the Comments: we do our best improve the language of the manuscript. Some sentence are broke down and simplify, so that it will more easy to understand.

06. Comments: 

Page 17 section 1: Remove underline & merge paragraph one to three.

Answer of the Comments: The underline removed and the paragraphs are merged.

07. Comments: 

Page 19, section 2.1: Italic

Answer of the Comments: The scientific name of pumpkin changed into italic format.

08. Comments: 

Page 19, section 2.2: Author recommended to put reference.

Answer of the Comments: References are added.

09. Comments: 

Page 19, section 3.3: Check for word size.

Answer of the Comments: The word sizes are corrected.

10. Comments: 

Page 19, section 3.4.3: why bold

Answer of the Comments: The word format is corrected.

RESPONSE TO REVIEWERS

Reviewer:

Thank you for your positive perspective on the manuscript.

The comments of the reviewer:

 All recommendations have been accepted. I have recommended the article for publication. Only one comment: in chapter 3.4.3, first column, m mole/100g is in bold and should be in normal font.

Comments of the answer:

According to the suggestion of the reviewer, the text was corrected.

---

## [Decision Letter · Decision Letter 2]

12 Jun 2024

PONE-D-23-44103R2Enhancing Nutritional Value and Quality of Cookies through Pumpkin Peel and Seed Powder FortificationPLOS ONE

Dear Dr. Asaduzzaman,

Thank you for submitting your manuscript to PLOS ONE. After careful consideration, we feel that it has merit but does not fully meet PLOS ONE’s publication criteria as it currently stands. Therefore, we invite you to submit a revised version of the manuscript that addresses the points raised during the review process.

We look forward to receiving your revised manuscript.

Kind regards,

Syed Amir Ashraf

Academic Editor

PLOS ONE

Journal Requirements:

Additional Editor Comments:

Authors are recommended to carefully check the manuscript for grammatical and basic error and do the revision as suggested by the reviewers and submit it as per the due date.  

Reviewers' comments:

Reviewer's Responses to Questions

**Comments to the Author**

1. If the authors have adequately addressed your comments raised in a previous round of review and you feel that this manuscript is now acceptable for publication, you may indicate that here to bypass the “Comments to the Author” section, enter your conflict of interest statement in the “Confidential to Editor” section, and submit your "Accept" recommendation.

Reviewer #4: (No Response)

2. Is the manuscript technically sound, and do the data support the conclusions?

Reviewer #4: Yes

3. Has the statistical analysis been performed appropriately and rigorously? 

Reviewer #4: Yes

4. Have the authors made all data underlying the findings in their manuscript fully available?

Reviewer #4: Yes

5. Is the manuscript presented in an intelligible fashion and written in standard English?

Reviewer #4: Yes

6. Review Comments to the Author

**Reviewer #4: **This manuscript makes a valuable contribution to the field of circular economy, describing a complex research that provides new insights into the possibility of exploiting the nutritional and bioactive composition of pumpkin by-products, such as peels and seeds, in the confectionery sector.

The study extends the knowledge of incorporating different pumpkin by-products (peels and seeds) in different concentrations to develop value-added cookie and noodle formulas.

The introduction is coherent and adequately structured, the literature review has been critically followed in line with the current state of knowledge. The aim of the research is clearly stated, the objectives are realistic, well defined and achievable.

The study is well planned and the methods are well suited to achieve the proposed objectives. The research methodology is generally well detailed, but needs some improvement, in particular in terms of design and presentation of flowcharts, as follows:

The cookie recipe, including ingredient quantities, could be put into one table for all cookies formulated with pumpkin seeds and pumpkin seeds, and a single cookie-making flowchart could be included because the process is identical. Quantities of materials do not have to be listed in the chart.

The same for noodles, a table with the recipe and a flowchart for their preparation.

The labelling of the biscuit and noodle samples respectively will be included in the tables in which the recipes were presented, so the list of samples will no longer be presented separately.

The part below will be removed:

SA = PA = Control (100% market wheat flour, Cookie)

SB = 95% market wheat flour /5%pumpkin seed cookie

SC = 90% market wheat flour /10%pumpkin seed cookie

SD = 85% market wheat flour /15%pumpkin seed cookie

SE = 80%market wheat flour /20%pumpkin seed cookie

PB = 95%wheat/5%pumpkin peel cookie

PC = 90%wheat/10%pumpkin peel cookie

PD = 80%wheat/20%pumpkin peel cookie

The list of the noodles samples are

NA = Control (100% wheat flour noodles)

NB = 90%wheat/10% pumpkin seeds noodles

NC = 90%wheat/10% pumpkin peels noodles

ND = 80%wheat/10%pumpkin seeds and 10% pumpkin peels noodles

Please pay more attention to the wording, e.g. “There were two types of cookies were formulated”, which could better be “Two types of cakes have been formulated.”

The results are clearly presented, but the number of 16 tables is excessive.

Tables 2, 3 and 4 could be merged (Proximate compositions of formulated cookies and noodles ), also Tables 7, 8 and 9 (Hunter Color Values (L*, a*,b*,ΔE) of formulated cookies and noodles), Tables 10, 11 and 12 (Total phenolic compound and antioxidant properties (DPPH and FRAP) of formulated cookies and noodles), Tables 15, 16 and 17 (Mean score for texture, aroma, color, taste and overall acceptability of formulated cookies and noodles).

The results obtained are discussed in detail and relevant explanations are provided, with the findings resulting from this research being well highlighted. This is a wide-ranging paper, which has generated many useful results of considerable practical value.

Some improvements are needed in the discussion to better highlight the added value of the research. Also, the RESULTS AND DISCUSSION section will be rearranged and renumbered accordingly.

The conclusions support the results of this study. This section needs to be improved, showing the genuine value and innovation of this study. What are the limitations of including pumpkin peel and seed powder in cookies and noodles?

The references are relevant to this research topic.

The study has high applicative potential, is a high quality and high impact approach in the field, therefore I recommend its publication after review.

7. PLOS authors have the option to publish the peer review history of their article (what does this mean?). If published, this will include your full peer review and any attached files.

Reviewer #4: No

---

## [Author Response · Author response to Decision Letter 2]

28 Jun 2024

RESPONSE TO EDITOR

01. Comments:

Answer of the Comments: We are extremely sorry; we cannot find any reference that should be retracted. We have reviewed all references, but none warrant retraction.

RESPONSE TO REVIEWERS

Reviewer:

Thank you for your positive perspective on the manuscript. We are pleased to hear that the introduction, methodology & references met your expectations. We will try to follow your suggestions to improve the study.

01) The comments of the reviewer:

The cookie recipe, including ingredient quantities, could be put into one table for all cookies formulated with pumpkin seeds and pumpkin seeds, and a single cookie-making flowchart could be included because the process is identical. Quantities of materials do not have to be listed in the chart.

The same for noodles, a table with the recipe and a flowchart for their preparation.

The labelling of the biscuit and noodle samples respectively will be included in the tables in which the recipes were presented, so the list of samples will no longer be presented separately.

Answer of the Comments: Following the reviewer's suggestion, we have included two tables in the study where all ingredient quantities are listed. The labeling has been added to the table, and the quantities have been removed from the chart. A flowchart has been prepared to represent all the cookies. Thank you for the excellent recommendation.

02) The comments of the reviewer:

The part below will be removed:

SA = PA = Control (100% market wheat flour, Cookie)

SB = 95% market wheat flour /5%pumpkin seed cookie

SC = 90% market wheat flour /10%pumpkin seed cookie

SD = 85% market wheat flour /15%pumpkin seed cookie

SE = 80%market wheat flour /20%pumpkin seed cookie

PB = 95%wheat/5%pumpkin peel cookie

PC = 90%wheat/10%pumpkin peel cookie

PD = 80%wheat/20%pumpkin peel cookie

The list of the noodles samples are

NA = Control (100% wheat flour noodles)

NB = 90%wheat/10% pumpkin seeds noodles

NC = 90%wheat/10% pumpkin peels noodles

ND = 80%wheat/10%pumpkin seeds and 10% pumpkin peels noodles

Answer of the Comments: According to the suggestion of the reviewer, this is removed.

03) The comments of the reviewer:

Please pay more attention to the wording, e.g. “There were two types of cookies were formulated”, which could better be “Two types of cakes have been formulated.”

Answer of the Comments: According to the suggestion of the reviewer, this is corrected.

04) The comments of the reviewer:

The results are clearly presented, but the number of 16 tables is excessive.

Tables 2, 3 and 4 could be merged (Proximate compositions of formulated cookies and noodles ), also Tables 7, 8 and 9 (Hunter Color Values (L*, a*,b*,ΔE) of formulated cookies and noodles), Tables 10, 11 and 12 (Total phenolic compound and antioxidant properties (DPPH and FRAP) of formulated cookies and noodles), Tables 15, 16 and 17 (Mean score for texture, aroma, color, taste and overall acceptability of formulated cookies and noodles).

Answer of the Comments: Following the reviewer's suggestion, Tables 2, 3, and 4 can be merged, as well as Tables 7, 8, and 9; Tables 10, 11, and 12; and Tables 15, 16, and 17. However, the statistical analysis differs for each table, and merging these tables would complicate the understanding of the sample differences. Nonetheless, if the reviewer recommends merging the tables, we will proceed accordingly.

05) The comments of the reviewer:

The results obtained are discussed in detail and relevant explanations are provided, with the findings resulting from this research being well highlighted. This is a wide-ranging paper, which has generated many useful results of considerable practical value.

Some improvements are needed in the discussion to better highlight the added value of the research. Also, the RESULTS AND DISCUSSION section will be rearranged and renumbered accordingly.

Answer of the Comments: 

We are pleased to hear that the result and discussion section met your expectations. All tables are rearranged and renumbered accordingly.

06) The comments of the reviewer:

The conclusions support the results of this study. This section needs to be improved, showing the genuine value and innovation of this study. What are the limitations of including pumpkin peel and seed powder in cookies and noodles?

Answer of the Comments: Thank you for your comments. This section now includes some important values as well as the limitations.

---

## [Editor Report · Decision Letter 3]

8 Jul 2024

Enhancing Nutritional Value and Quality of Cookies through Pumpkin Peel and Seed Powder Fortification

PONE-D-23-44103R3

Dear Dr. Asaduzzaman,

We’re pleased to inform you that your manuscript has been judged scientifically suitable for publication and will be formally accepted for publication once it meets all outstanding technical requirements.

Kind regards,

Syed Amir Ashraf

Academic Editor

PLOS ONE
---

## [Editor Report · Acceptance letter]

11 Jul 2024

PONE-D-23-44103R3 

PLOS ONE

Dear Dr. Asaduzzaman, 

I'm pleased to inform you that your manuscript has been deemed suitable for publication in PLOS ONE. Congratulations! Your manuscript is now being handed over to our production team.

Kind regards, 

on behalf of

Dr. Syed Amir Ashraf 

Academic Editor

PLOS ONE